# Genomics Analysis Reveals the Potential Biocontrol Mechanism of *Pseudomonas aeruginosa* QY43 against *Fusarium pseudograminearum*

**DOI:** 10.3390/jof10040298

**Published:** 2024-04-21

**Authors:** Jiaxing Meng, Feifei Zan, Zheran Liu, Yuan Zhang, Cancan Qin, Lingjun Hao, Zhifang Wang, Limin Wang, Dongmei Liu, Shen Liang, Honglian Li, Haiyang Li, Shengli Ding

**Affiliations:** 1College of Plant Protection, Henan Agricultural University, Zhengzhou 450046, China; mjx135207@163.com (J.M.); zff1522695@163.com (F.Z.); lzr19982021@163.com (Z.L.); 17837184209@163.com (Y.Z.); a18839327647@163.com (C.Q.); 13526896725@163.com (L.H.); meretals@126.com (Z.W.); limin-wang@foxmail.com (L.W.); honglianli@sina.com (H.L.); 2Institute of Quality Standards and Testing Technology for Agro-Products, Henan Academy of Agricultural Sciences, Zhengzhou 450002, China; dongmeiliu80@163.com; 3Horticulture Research Institute, Henan Academy of Agricultural Sciences, Zhengzhou 450002, China; liangshen1@126.com; 4National Key Laboratory of Wheat and Maize Crop Science, Zhengzhou 450046, China

**Keywords:** *Fusarium* crown rot, *Fusarium pseudograminearum*, biocontrol bacteria, *Pseudomonas aeruginosa*, genomics, biocontrol factors

## Abstract

*Fusarium* crown rot (FCR) in wheat is a prevalent soil-borne disease worldwide and poses a significant threat to the production of wheat (*Triticum aestivum*) in China, with *F. pseudograminearum* being the dominant pathogen. Currently, there is a shortage of biocontrol resources to control FCR induced by *F. pseudograminearum*, along with biocontrol mechanisms. In this study, we have identified 37 strains of biocontrol bacteria displaying antagonistic effects against *F. pseudograminearum* from over 8000 single colonies isolated from soil samples with a high incidence of FCR. Among them, QY43 exhibited remarkable efficacy in controlling FCR. Further analysis identified the isolate QY43 as *Pseudomonas aeruginosa*, based on its colony morphology and molecular biology. In vitro, QY43 significantly inhibited the growth, conidial germination, and the pathogenicity of *F. pseudograminearum*. In addition, QY43 exhibited a broad spectrum of antagonistic activities against several plant pathogens. The genomics analysis revealed that there are genes encoding potential biocontrol factors in the genome of QY43. The experimental results confirmed that QY43 secretes biocontrol factor siderophores and pyocyanin. In summary, QY43 exhibits a broad spectrum of antagonistic activities and the capacity to produce diverse biocontrol factors, thereby showing substantial potential for biocontrol applications to plant disease.

## 1. Introduction

Common wheat (*Triticum aestivum*) is one of the most important cereal crops. China is the largest wheat-producing and -consuming country, and it accounted for approximately 17.5% of the global wheat production from 2011 to 2020 [1]. In 2023, the wheat grain output reached approximately 20% of the total grain output of China. Therefore, wheat plays a crucial role in ensuring China’s food security. The prevention and control of wheat diseases is of great significance to the yield and quality of wheat. *Fusarium* crown rot (FCR) is a soil-borne disease caused by several *Fusarium* spp. strains occurring in many wheat-producing areas worldwide and has a serious impact on the quality and yield of wheat [2,3]. *F. pseudograminearum*, *F. graminearum*, and *F. culmorum* are the main pathogens responsible for FCR [4]. In China, our research group first reported FCR caused by *F. pseudograminearum* [5]. It has been found that the distribution of pathogenic fungi contributing to FCR is relatively complex, and *F. pseudograminearum* is the dominant pathogen of FCR [6]. Additionally, deoxynivalenol (DON), nivalenol (NIV), and other mycotoxins produced by *F. pseudograminearum* cause severe threats to food security [7]. Above all, the incidence area of FCR is increasing annually, which is becoming a major problem in agricultural production.

In line with the policy of “prevention first and comprehensive control” in China, the approach to preventing and controlling FCR encompasses a combination of diverse control methods. Currently, the primary methods of managing this disease include chemical and biological control. Chemical control measures can provide the rapid and effective management of FCR. Early-stage applications of pyraclostrobin, phenamacril, fludioxonil, tebuconazole, and carbendazim sprays have demonstrated good control effects against FCR [8,9,10]. Although chemical fungicides constitute a rapid and effective management method, they result in environmental pollution, hazard to human health, and pesticide residues [11].

Over the last 50 years, many breakthroughs involving biological control have been made in plant disease control [12]. Biological control involves harnessing microorganisms, or their secondary metabolites, to use against pathogens. Biocontrol bacteria can inhibit the growth of pathogenic fungi and reduce the occurrence of diseases, with a small impact on the environment. As FCR is becoming increasingly serious in China, safe and reliable biological control agents have become urgently needed. As reported, *Trichoderma harzianum* has been shown to effectively control FCR, as evidenced by Kthiri et al. [13]. Additionally, *Bacillus siamensis* YB-1631 appears to have significant potential in promoting wheat growth and controlling wheat FCR [14]. Besides *Pseudomonas fluorescens*, *P. putida*, and *P. aeruginosa* in the *Pseudomonas* clade are important resources for developing biological control agents [15]. Over the following two decades, dozens of members of the *P. aeruginosa* species have been isolated and identified as biocontrol strains [16,17,18,19,20,21]; however, there have been no reports on the biocontrol application of *P. aeruginosa* against FCR.

Whole-genome sequencing is pivotal for analyzing the complete set of genes within an organism and is crucial for understanding the composition, structure, and function of the genome. Advancements in the genomic analysis of biocontrol bacteria have provided an efficient method of rapid identification of biocontrol factors. *P. aeruginosa* can manage *F. graminearum* by producing diverse antifungal compounds [19,22]. To gain further insights into the genes associated with biocontrol factors in the genomes of biocontrol bacteria, a comparative genomics approach was used to refer to the similarities between the known strain genomes. A further comparative genomics study was carried out between biocontrol bacteria *P*. *aeruginosa* RK1 and UCBPP-PA14, and it revealed several biocontrol factors and plant-growth-promoting related genes [21]. Through a comparative genomics analysis of *P. aeruginosa* NG61 with PAO1, Rikame and Borde analyzed the gene related to plant-growth-promoting traits [23]. Through comparative genomics studies, specific gene clusters related to the biocontrol of biocontrol bacteria can be discovered.

In this current study, to identify the biocontrol resources capable of controlling FCR in wheat, we screened the biocontrol bacteria *P. aeruginosa* QY43 with an inhibitory effect on the growth of *F. pseudograminearum*. The genomic analysis of QY43 predicted the biocontrol factors and the specific polysaccharide antibiotics. In addition, the experiment demonstrated that QY43 secretes biocontrol factors siderophores and pyocyanin. This study provides biological control materials and a theoretical basis for the biocontrol mechanism of *P. aeruginosa* QY43 against FCR.

## 2. Materials and Methods

### 2.1. Sample Collection and Bacterial Isolation

Three soil samples were taken from the field soil after wheat harvest in Qinyang City (112.986347° E, 35.052226° N), Henan Province, China, and stored in the laboratory at the Henan Agricultural University in the summer of 2021. Using the dilution-plate method, 100 µL of one-thousand-fold diluted suspension was spread onto tryptic soy broth (TSB) plates (17 g tryptone, 3 g soy peptone, 2.5 g glucose, 5.0 g NaCl, 2.5 g KH_2_PO_4_, and 20 g agar to prepare 1 L of medium with distilled water; pH = 7.3 ± 0.2) and incubated for 1 d at 25 °C. A single colony was then isolated from each TSB plate. 

### 2.2. Screening and Identification of Antagonistic Strains In Vitro

The isolate WZ-8A was isolated from the diseased stem bases with brown discoloration of wheat cultivar Aikang 58 in 2012 in Wuzhi county (WZ: abbreviation for the isolates), Henan Province, China, confirmed by the morphological and molecular methods as *F. pseudograminearum*. This strain has a strong pathogenicity and a relatively stable phenotype on Aikang58, which was to be used as a standard strain by our research group [24,25]. The WZ-8A was deposited in the Agriculture Culture Collection of China with No. ACCC 38068.

The initial screening to assess the antagonistic activity of the bacterial strains against WZ-8A was performed on potato dextrose agar (PDA) plates (200 g potato, 20 g dextrose, and 20 g agar to prepare 1 L of medium with distilled water). A 5-mm mycelial plug of WZ-8A, taken from the fresh edge of the colony, was placed in the center of a 9-cm Petri dish containing 15 mL of PDA medium. The single bacterial colony was inoculated 2.5 cm from the plug. Eight different single colonies were inoculated into each plate and incubated at 25 °C for 3 d in the dark. This experiment was performed one time, with one replicate set up. The bacteria that exhibited antagonistic effects against WZ-8A were selected for subsequent testing.

To explore the antagonistic activities of QY43 against WZ-8A, two drops (1 µL) of bacterial solution of the QY43 cell suspension, cultured for 24 h in nutrient broth medium (NB medium) [26] (5 g peptone, 3 g beef extract, 5 g NaCl, and 1 L of distilled water; pH = 7.4 ± 0.1), were deposited around the plug, approximately 2.5 cm away. The plates were stored at 25 °C for 3 d in the dark. The diameter of each colony on the plates was measured in two perpendicular directions. The formula is as follows: antagonistic activity (%) = (control colony diameter − treatment colony diameter)/(control colony diameter − mycelial plug diameter).

The QY43 strain was identified using 16S ribosomal DNA (rDNA) sequencing and phenotype analysis. Genomic DNA was extracted using a bacterial genomic DNA extraction kit (Sangon Biotech (Shanghai) Co., Ltd., Shanghai, China). The 16S rDNA fragment was subjected to amplification via PCR using the primer pair 27F (5′-AGAGTTTGATCCTGGCTCAG-3′)/1492R (5′-TACGGCTACCTTGTTACGACTT-3′) and was sequenced by Sangon Biotech (Shanghai) Co., Ltd. Using BLAST (https://blast.ncbi.nlm.nih.gov/Blast.cgi, (accessed on 11 June 2023)), the 16S rDNA sequence of QY43 was compared with sequences from the National Center for Biotechnology Information (NCBI) database, and a relevant phylogenetic tree was constructed.

### 2.3. Antagonistic Activity of QY43 against Conidia Germination of F. pseudograminearum

The QY43 was cultured in NB medium at 25 °C for 24 h in a shaker. The QY43 was introduced into the NB medium to achieve an optical density (OD) 600 value of 0.1. For the conidiation of WZ-8A, fresh culture plugs were transferred from PDA plates into 150-mL flasks containing 100 mL of carboxymethyl cellulose (CMC) medium (NH_4_NO_3_ 1 g, KH_2_PO_4_ 1 g, MgSO_4_·7H_2_O 0.5 g, yeast extract 1 g, CMC-Na 15 g, and 1 L of medium with distilled water), with shaking at 150 rpm at 25 °C for 4 d. The conidia of WZ-8A were standardized to a concentration of 1 × 10^6^ conidia/mL using the dilution counting method under a microscope. Adjusted concentrations of the bacterial solution of QY43 and conidial suspension of WZ-8A were mixed in a 1:1 ratio. Simultaneously, *Escherichia coli* DH5α was cultured under the same conditions as the control group. The germination of conidia is considered to have occurred when the length of the tube exceeds 2/3 of the length of the conidia. After incubating the conidial suspensions at 25 °C for 6 h, the percentage of germinated conidia was assessed under a microscope (Axio image M2 microscope, Carl Zeiss, Germany). Observations were made on the effect of different treatments on the mycelium of *F. pseudograminearum* after 12 h of incubating.

### 2.4. Efficacy of QY43 against FCR in a Glasshouse

Four mycelium plugs of *F. pseudograminearum* were incubated in the sterilized millets and kept in the dark at 25 °C for 7 d, shaking the bottles once a day. A pot experiment was performed by introducing sterilized soil inoculated with 0.5% millet containing *F. pseudograminearum* along with pregerminated wheat seeds, which were soaked for 2 h in a bacterial solution of QY43 with OD600 = 0.8 before growing. Twenty-one days after planting, the disease of the wheat was observed and recorded using a rating scale from 0 to 7 [27], where 0 = no disease in the plant; 1 = the first leaf sheath was slightly brown or the ground stem was brown.; 2 = the first leaf sheath turned brown but did not turn black; 3 = the first leaf sheath became black or the second leaf sheath became brown; 4 = the third leaf sheath turned brown or the wheat grew slowly; 5 = dying plant with rotted culm base; 6 = dead plant with rotted stem base (plants had matured to produce more than three leaves); and 7 = plant death by rotting of the seed or by pre-emergence damping-off of the seedling, following the equation: disease index = (total number of tillers − number of disease severity class 0 tillers)/total number of tillers × 100. Then, the wheat was removed from the pot, the soil on the root was washed away, and the wheat was spread on a white plate for photos to observe the symptoms of FCR and the control effect of QY43.

### 2.5. Assessment of Broad-Spectrum Antagonistic Activity of QY43

The strains tested in this study were *F. graminearum* PH-1, *Bipolaris sorokiniana* LK93, *Rhizoctonia zeae* R0301, *Magnaporthe oryzae* RB22, *Phytophthora capsici* LT263, *Botryosphaeria dothidea* I5, and *Valsa mali* A3. The strains tested in this study were acquired from other researchers in related fields and had been stored in the laboratory at the Henan Agricultural University. *P. capsici* was cultivated on V8 medium (V8 Juice 100 mL, CaCO_3_ 1 g, and 18 g agar to prepare 1 L of medium with distilled water), while the other strains were cultured on PDA plates. A 5-mm plug of the test strain was centrally inoculated into the culture medium, and the bacterial solution of QY43 was inoculated 2.5 cm away from the plug. The plate cultures were incubated at 25 °C for 3 d in the dark. The diameter of each colony on the plates was measured in two perpendicular directions.

### 2.6. Whole-Genome Sequencing of QY43

To identify the possible antifungal compounds produced by QY43, the whole genome of QY43 was sequenced by BGI Co., Ltd., (Wuhan, China). The overall analysis can be divided into four modules, as follows: (1) Data filtering: the raw data were obtained through PacBio Sequel II (Pacbio, Menlo Park, CA, USA), and then the clean data were generated after filtering. (2) Assembly: assembly of the reads after filtering into the genome and assessing the assembly with the software proovread (version 2.12) and GATK (version v1.6-13). (3) Genomic component analysis with the software Glimmer (version 3.02) [28], including (a) analysis of repeat sequences with the software Tandem Repeat Finder (version 4.04) [29]; (b) CRISPER prediction with the software CRISPRCasFinder (version 4.2.19) [30]; (c) non-coding RNA prediction, including rRNA with the software RNAmmer (version 1.2) [31], tRNA with the software tRNAscan-SE (version 1.3.1) [32], and sRNA with the software Rfam (version 9.1) [33]; (d) prophage prediction with the software PhiSpy (version 3.7.8) [34]. (4) Analysis of gene function, including the analysis of animal pathogens, including virulence factor database (VFDB) annotation [35].

### 2.7. Comparative Genomics Analysis to Identify Genes Involved in the Synthesis of Antifungal Compounds Specific to QY43

In this study, a collinearity analysis was conducted at the nucleotide level between QY43 and four strains of *P. aeruginosa*, as follows: PUPa3, PGPR2, ID4365, and M18 were analyzed with the software MUMmer (version 3.22) [36]. In the present study, we compared the gene families of the genome of QY43 with those of different *P. aeruginosa* strains. The comparison involved analyzing the gene sequences of QY43 against those of the four specified *P. aeruginosa* strains and assessing the number of shared and distinct genes between them, which were analyzed with the software CD-HIT (version v4.6.6) [37]. Based on the strain core gene, a phylogenetic tree was constructed using TreeBeST (version treebest-1.9.2) [38].

### 2.8. Characterization of the Biological Properties and Antifungal Compounds of QY43 In Vitro

Phosphate solubilization test: Phosphate solubilization was evaluated using the halo zone method. Biocontrol bacteria were inoculated at the center of Pikovskaya (PKO) inorganic medium (10 g glucose, 2.5 g Ca_3_(PO_4_)_2_, 0.5 g (NH_4_)_2_SO_4_, 0.2 g NaCl, 0.2 g KCl, 0.03 g MgSO_4_·7H_2_O, 0.003 g FeSO_4_·7H_2_O, 0.03 g MnSO_4_·4H_2_O, 0.4 g yeast extract, and 18 g agar to prepare 1 L of medium with distilled water) and incubated at 30 °C for 72 h [39]. The size of the phosphate solubilization halo was observed.

Cellulose degradation ability test: The plate clear-zone method was used. The biocontrol bacteria were point-inoculated into cellulose degradation detection medium (10 g peptone, 10 g yeast extract, 10 g carboxymethylcellulose, 1 g KH_2_PO_4_, 5 g NaCl, and 18 g agar to prepare 1 L of medium with distilled water) and incubated for 72 h at 30 °C [40]. Congo red staining was performed by soaking the culture plate with a 1 g L^−1^ Congo red solution for 1 h. After discarding the Congo red solution, 1 mol L^−1^ NaCl solution was added to the culture dish, which was then soaked for 1 h. The presence of clear zones indicated cellulose degradation.

Protease secretion ability test: The plate clear-zone method was used. The strains were point-inoculated into a protease identification medium (10 g gelatin and 18 g agar to prepare 1 L of medium with distilled water) and cultured for 72 h at 30 °C [41]. A mercuric chloride solution was added to the protease identification medium after incubation. Large clear zones indicated strong enzymatic activity.

Amylase secretion ability test: The plate clear-zone method was used. The strains were point-inoculated into an amylase identification medium (1 g soluble starch, 5 g peptone, 5 g glucose, 5 g NaCl, 5 g beef extract, and 18 g agar to prepare 1 L of medium with distilled water) and cultured for 72 h at 30 °C [42]. After incubation, an iodine solution was added to the amylase identification medium to observe the presence of clear zones. Large clear zones indicated strong enzymatic activity.

Siderophore secretion ability test: The plate clear-zone method was used. The strains were inoculated into the chrome azurol S (CAS) detection medium (product code: HB9132; Hopebio Biological Company) [43]. The powder (10.87 g) was dissolved in 1000 mL of distilled water, heated to complete dissolution, and incubated at 30 °C for 48–72 h. The observation of clear zones around the colonies indicated siderophore production.

Biofilm formation ability test: Crystal violet staining is the most used chemical method for the quantitative detection of bacterial biofilms [44,45,46]. This method was used to identify whether QY43 had the ability to generate bacterial biofilms. Fresh bacterial solution (50 μL) was inoculated into 450 μL of NB medium in a 1.5-mL centrifuge tube and incubated statically at 30 °C for 24 h. After removing the bacterial solution, the tubes were gently rinsed with sterilized water and stained with 1% crystal violet for 15 min. Subsequently, the crystal violet solution was removed, and the tube was washed with sterilized water. Residual purple staining of the tube walls indicated biofilm formation. All of the experiments above were performed two times independently, with three replicates each time.

To prepare the cell-free supernatant, a bacterial solution of QY43 (OD600 = 0.1) was utilized, following the previously described method in Section 2.4. A total of 1 mL of this solution was added to 150 mL of fermentation medium (5 g sucrose, 10 g tryptone, 2.5 g NaCl, and 1 L distilled water adjusted to pH 7.5). QY43 was then cultivated in this fermentation medium at 25 °C at 150 rpm for 2 d. The supernatant of QY43 was collected and then filtered through a 0.22-µm bacterial filter (product code: FPV203030 Guangzhou Jet Bio-Filtration Co., Ltd., Guangzhou, China) to obtain the sterilized fermentation broth. Then, the *F. pseudograminearum* was incubated on PDA with diluted to five and ten times that of the original volume.

To investigate the thermal stability of the antifungal compounds in the fermentation broth of QY43, they were treated at 40, 60, 80, and 121 °C for 20 min, respectively. After cooling, the PDA was mixed at a ratio of 1:9, and the WZ-8A was inoculated in the center of the plate. The diameter was measured after incubating at 25 °C for 3 d, the growth inhibition rate of the *F. pseudograminearum* was calculated, and the different temperatures were observed.

Additionally, after obtaining the fermentation broth following the same procedure as that mentioned above, CH_3_OH was added in a three-fold volume. The mixture was allowed to precipitate overnight at 4 °C and then centrifuged at 10,000 rpm for 20 min at 4 °C to obtain the peptide-rich precipitate. The supernatant was evaporated at 45 °C to remove the methanol and subsequently diluted to the original concentration with phosphate buffer saline (PBS), which was used in the following experiment. After obtaining the fermentation broth following the procedure mentioned above, (NH_4_)_2_SO_4_ was added to the saturation and precipitated overnight at 4 °C. Subsequently, the mixture was centrifuged at 6000 rpm for 30 min at 4 °C. The obtained precipitate was dissolved in PBS and then dialyzed at 4 °C for 6 h, with the PBS being changed every 2 h. Following dialysis, the PBS containing peptides was filtered through a 0.22-μm bacterial filter to obtain a sterilized peptide solution. The sterilized polypeptide solution was combined with the PDA medium at a 1:9 ratio. As a control, PBS was mixed with PDA at a ratio of 1:9. After cooling, a 5-mm plug of WZ-8A was inoculated and placed in a 25-°C incubator for 3 d in the dark.

In the experiment of the detection of pyocyanin, the detection medium for pyocyanin, as described (20 g peptone, 1.4 g MgCl_2_, 10 g K_2_SO_4_, and agar 20 g to prepare 1 L of medium with distilled water; pH = 7.4 ± 0.1) [47], was prepared and inoculated with 20 μL of the bacterial solution (OD600 = 0.1), with incubation at 25 °C for 24 h. After breaking up the medium, 3–5 mL of chloroform was added to each tube, followed by vigorous shaking to transfer the pigment from the medium to chloroform. Subsequently, the chloroform was transferred to a new tube and 1 mL of 1 M HCl was added. If the pigment contained pyocyanin, the upper portion of the HCl solution in the tube exhibited a red coloration. This experiment was performed two times independently, with three replicates being set up each time.

### 2.9. Statistical Analysis

All experiments were performed at least three times independently, with three replicates each time, except those specifically noted in each section. Standard deviation (s.d.) and Student’s *t* test were used to analyze the data with the software Statistical Product and Service (SPSS; version 20.0; IBM, Armonk, NY, USA). The significant difference analysis was conducted, respectively, between the different treatment groups alone and the control group. The results of the Student’s *t* test were as follows: *p* < 0.05 indicated a significant difference between treatments, while *p* < 0.01 indicated an extremely significant difference between treatments.

## 3. Results

### 3.1. QY43 Was Identified as P. aeruginosa with Biocontrol Ability through Screening

To identify biocontrol bacteria against FCR, we collected soil samples from wheat fields with a high incidence of FCR in Qinyang city, Henan Province. Using the dilution-plate method, we isolated over 8000 single colonies from these soil samples. Through the experiment, a total of 37 isolates were identified with antagonistic activity against *F. pseudograminearum* (Appendix A). Among these strains, QY43 exhibited the highest antagonistic efficacy, thereby meriting further investigation (Figure 1A,B). Morphologically, QY43 displayed a smooth, moist colony with irregular edges, featuring a flat, white luster when cultivated on NA medium (Figure 1C). The Gram staining microscopic observations revealed QY43 to be a Gram-negative strain, which has a short, rod-shaped red staining morphology. (Figure 1D) [48].

To further identify the biocontrol bacteria QY43, we revealed 100% sequence similarity to the 16S rDNA sequence of *P. aeruginosa* strain, D1-3 (MN922571). The construction of a phylogenetic tree based on the 16S rDNA sequences of QY43 and various sequenced *Pseudomonas spp.* from NCBI GenBank revealed that the closest relative to QY43 was *P. aeruginosa* D1-3 (Figure 1E). Therefore, based on the analysis of the colony morphology and the evolutionary relationship, QY43 was confirmed to belong to *P. aeruginosa*. 

### 3.2. QY43 Inhibits Conidial Germination, Causes Plasmolysis, and Reduces Pathogenicity of F. pseudograminearum

To further evaluate the antagonistic activity of QY43 against WZ-8A in vitro, we determined the effects of QY43 on the conidial germination of WZ-8A. After co-culturing with the bacterial solution of QY43 for 6 h, the conidial germination of WZ-8A exhibited a substantial inhibition by approximately 60% compared to that of the CK (Figure 2A,B). After co-culturing with the bacterial solution of QY43 for 12 h, the conidia of WZ-8A showed growth and developmental abnormalities, in that the conidia of WZ-8A appear plasmolysis with co-culturing with the bacterial solution of QY43 (Figure 2C). At the same time, the conidia of WZ-8A growth and development were normal in the CK and DH5α treatment groups.

To assess the efficacy of QY43 against FCR, an indoor pot experiment was conducted. The disease index of FCR was investigated at 21 d after sowing the QY43-soaked seeds and the non-soaked ones (Figure 3A), the disease index was significantly reduced (*p* < 0.05) (Figure 3B) and the harm of FCR to the plant stem base and root was reduced after treatment with QY43. Compared to the wheat that was not treated with the QY43 bacterial solution, those treated with the QY43 bacterial solution exhibited a longer root length and more vigorous aboveground growth (Figure 3C). The QY43 bacterial solution immersion mitigated the damage caused by FCR to the wheat. These findings indicated a substantial control effect of QY43 on FCR (Figure 3).

### 3.3. QY43 Exhibited Broad-Spectrum Antagonistic Activity against Plant Pathogens

To further explore the biocontrol potential of QY43, its broad-spectrum antagonistic activity was extensively investigated. QY43 was evaluated for its antagonistic effects against a range of plant pathogens, including *F. graminearum* PH-1, *M. oryzae* RB22, *R. zeae* R0301, *B. sorokiniana* LK93, *P. capsici* LT263, *B. dothidea* I5, and *V. mali* A3 (Figure 4A). The assessment revealed substantial antagonistic activities of QY43 against these pathogens, reaching 78.74%, 80.81%, 54.61%, 73.67%, 66.03%, 65%, and 85.39% for PH-1, LK93, RB22, LT263, R0301, A3, and I5, respectively (Figure 4B). Therefore, QY43 exhibits broad-spectrum antagonistic activity, suggesting its potential as a viable biological control agent against plant pathogens, including fungi and oomycetes.

### 3.4. The Potential Biocontrol Factors of QY43 Were Identified through Genome Analysis

The DNA of QY43 was sequenced and analyzed through PacBio Sequel II, in which the low-quality and splice sequences contained in the original sequencing data were removed. Hence, the data processing and monitoring showed that the data were reliable for the subsequent analysis (Appendix A). The results showed that the genome size of QY43 was 647,7906 bp, with 6079 genes; the total gene length was 577,1946 bp, with the average length being 949.49 bp; the CG content of the genome of QY43 was 67.09%; and the proportion of gene length to the total genome length was 89.1%. The gene length distributions are shown in Appendix A. Based on the GC skew analysis and the distributions of genes and ncRNA, the gene annotation Circos software (version 0.69-7, Vancouver, British Columbia, Canada) was used to display the circular genome map of QY43, as shown in Figure 5. The genome analysis of QY43 revealed that there were 682 potential biocontrol factors, indicating that QY43 is capable of effectively inhibiting plant pathogen growth (Appendix A). The genome of QY43 contained 19 phz-related genes related to the production of phenazine-1-carboxamide acid, phenazine-1-carboxamide, 1-hydroxyphenazine, and pyocyanin. Meanwhile five genes (*pvdA*, *pvdD*, *pvdF*, *pvdI*, and *pvdJ*) are likely to be directly responsible for the synthesis of pyocyanin polypeptides in the genome of QY43 [50]. Chorismic acid is converted into salicylic acid under the catalytic action of *pchA* and *pchB* in the genome of QY43. Subsequently, through the catalysis of *pchC*, *pchD*, and *pchEFGHI*, salicylic acid was transformed into siderophores by QY43 [51]. Furthermore, there are 5 proteins/enzymes (RhlA, RhlB, RhlC, RhlG, and RhlI) that are critical for rhamnolipid production [52], 22 genes involving the biosynthesis of exopolysaccharides, and 14 biofilm-formation-related genes. Moreover, QY43 had numerous genomic signatures that favor plant growth, including two genes (*phoQ* and *phoP*) involved in phosphate metabolism [53]. In summary, QY43 demonstrates promising potential for a wide range of applications in biocontrol and plant growth promotion [54].

### 3.5. Comparative Genomics Revealed That QY43 Has Three Unique Genes Related to the Synthesis of Polysaccharide Antifungal Compounds

Based on comparative genomic and nucleotide similarities (Appendix A), we identified unique and common genes in the genomes of QY43 and the following four reported biocontrol bacteria: PUPa3, PGPR2, ID4365, and M18 (Figure 6A). According to the core gene analysis with the genome of the five strains, QY43 and PUPa3 displayed the highest homology with QY43 (Figure 6B). However, there are 252 unique genes that exist in the genome of QY43. By conducting a combined analysis of the VFDB database, we identified three unique biosynthetic genes of polysaccharide antifungal compound that are related to biocontrol (Table 1). Therefore, it is predicted that QY43 can produce specific polysaccharide antibiotics.

### 3.6. Biocontrol Factors of P. aeruginosa QY43 Were Detected by the Experiment

To delve deeper into the antifungal compound of QY43, we assessed its physical and chemical properties. These findings demonstrate that QY43 exhibited the secretion of siderophore, the degradation of cellulose, protein, and inorganic phosphorus, and the generation of biofilm (Table 2).

The fermentation broth inhibited the growth of WZ-8A by 93.33% and 97.33% (Figure 7A,B). To explore the stability of the antifungal compounds in the fermentation broth, the different temperatures of the fermentation broth were tested. In addition, the inhibitory effect on WZ-8A decreased significantly with the increasing temperature (Figure 7C,D). To further identify the antifungal compounds produced by QY43, the antagonistic activity of the QY43 polypeptides obtained through (NH_4_)_2_SO_4_ precipitation and CH_3_OH extraction were tested (Figure 7E). It was experimentally discovered that the peptides extracted using the above two methods have certain antifungal activity (Figure 7F). The results of the genomic analysis indicated that QY43 may produce pyocyanin, which was also detected with color reaction (Figure 7G). Based on the analysis of the characteristics of QY43, it was established to be capable of forming biofilms and secreting antifungal compounds, such as siderophores and pyocyanin. In addition, the strain may have the potential to promote plant growth (Table 2). Thus, QY43 could have diverse practical agricultural applications as a biological control agent and biofertilizer.

## 4. Discussion

Biocontrol represents a valuable tool in sustainable agriculture, offering effective plant disease management solutions that prioritize environmental protection, human health, and long-term agricultural sustainability. In this study, we screened *P. aeruginosa* QY43 as a biocontrol bacterium, which displayed very significant antagonistic effects against *F. pseudograminearum*. QY43 also exhibited broad-spectrum antagonistic activity against other plant pathogenic fungi. The genomic analysis revealed genes encoding potential biocontrol factors, corroborated by experimental evidence of biocontrol factor siderophore secretion, suggesting the promotion of plant growth as an indirect mechanism to reduce disease index.

Biocontrol factors possess mechanisms that operate either directly or indirectly to combat plant diseases or promote plant growth. *P. aeruginosa*, a conditional pathogen, causes infections in both animals and humans [55]. However, a series of rhizosphere-derived *P. aeruginosa* strains with remarkable biocontrol efficacy have been isolated and identified in recent years [16,17,18,19,20,21,56]. These strains can produce various metabolites that antagonize plant pathogens. We have also further explored the possible biocontrol factors produced by *P. aeruginosa* QY43. Based on the genome analysis of QY43, we have identified some key biosynthetic pathways of the biocontrol factors. For example, phenazine antibiotics, pyoverdine and pyochelin, rhamnolipids, and exopolysaccharides are predicted to be synthesized by QY43. Phenazine antibiotics produced by numerous *Pseudomonas* spp. of biocontrol bacteria inhibited the growth of plant pathogenic fungi [57,58]. The siderophores produced by *P. aeruginosa* biocontrol bacteria, which are known to be involved in antifungal activity, as well as plant growth and development [59]. Rhamnolipids are a class of biosurfactants with great potential for industrial use [52]. Furthermore, though experimentation, we verified that QY43 secretes antifungal compounds, such as siderophores and pyocyanin. Our experiments show that QY43 has great biocontrol potential. A comparative genomic analysis of the four known biocontrol strains indicated that QY43 could produce antifungal compounds like the reference strains, for instance, phenazine-1-carboxamide acid, siderophores, pyochelin, salicylic acid, and rhamnolipids [59,60,61,62]; however, there are also three unique biocontrol polysaccharide antifungal compound synthesis genes specific to QY43. LPS is a major biocontrol factor, which can induce plant immunity and improve crop resistance [63,64]. The O-antigen of LPS polymerization length is affected by the interaction of Wzz (QY43GL001843) with Wzy [65]. WbpA (QY43GL001844) of *P. aeruginosa* is essential for O-antigen biosynthesis. CsrA (QY43GL004463) is a well-known global post-transcriptional regulator that is critical for controlling the production of LPS by *P. aeruginosa* in response to extracellular stimuli [66,67]. Comparative genomics revealed that QY43 produces specific polysaccharide antifungal compounds, indicating significant research value and biological control potential.

This study did not specifically investigate the potential biocontrol mechanism that enhances crop resistance against FCR. However, QY43 showed extensive antagonistic activity, displaying inhibitory effects against various plant pathogens, suggesting its potential as a viable biological control agent against plant pathogens affecting wheat and various other plants. In China, *P. aeruginosa* M18 is a crucial industrial strain used to produce green pesticides (phenazine-1-carboxylic acid). By constructing a chromosomally non-scar triple-deleted mutant M18MSU1 and optimizing the culture medium, a phenazine-1-carboxamide acid production yield of 4.7712 g/L was achieved in validation experiments [68]. Future research will focus on further elucidating the mechanisms underlying QY43’s antagonistic activities, optimizing its application methods, and assessing its efficacy under field conditions, aiming to further apply it in agricultural production. In summary, QY43 showcases diverse antagonistic activities and biocontrol factor production, underscoring its promising potential for combating plant diseases through biocontrol.

## 5. Conclusions

In this study, we identified a novel *P. aeruginosa* strain, QY43, which exhibited substantial control effects against FCR and extensive antagonistic activity. Genomics analysis revealed that QY43 has the ability to secrete diverse biocontrol factors. Some of the biocontrol factors exhibited by QY43 have been identified as antifungal compounds, and others may be beneficial to plant growth. The secondary metabolites of QY43 revealed significant potential for a broad range of applications in agriculture.

## Figures and Tables

**Figure 1 jof-10-00298-f001:**
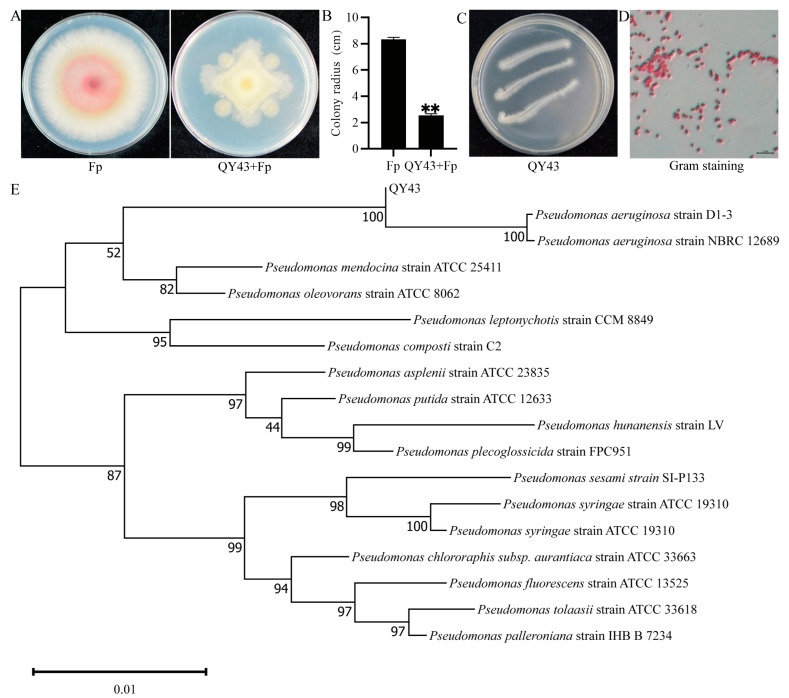
Identification of biocontrol bacteria QY43 against *Fusarium pseudograminearum*. (**A**): Colony morphology of *F. pseudograminearum* WZ-8A (Fp) on a PDA plate with and without QY43; (**B**): Colony radius of WZ-8A on a PDA plate and after co-culturing with QY43. The mean ± s.d. (standard deviation) from three independent experiments; ** indicates extremely significant differences at a *p*-value of 0.01. Bar = 5 µm; (**C**): Colony morphology of QY43 on an NA plate; (**D**): Gram staining of QY43; (**E**): Phylogenetic tree of QY43 and *Pseudomonas* spp. Phylogenetic analyses of the 16S rDNA sequences were performed using the neighbor-joining method with Molecular Evolutionary Genetics Analysis (version 7.0) [49].

**Figure 2 jof-10-00298-f002:**
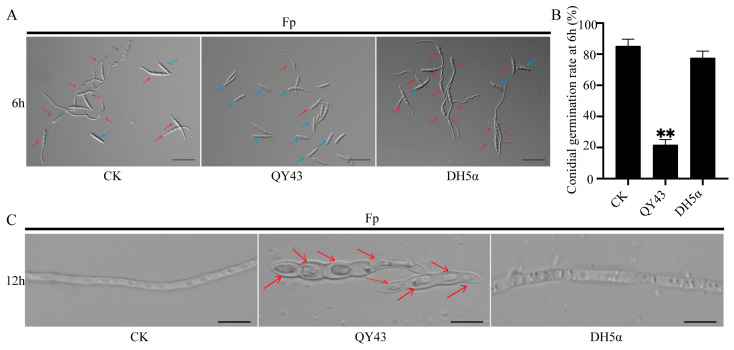
Effect of QY43 on the conidial germination and development of *F. pseudograminearum* WZ-8A (Fp). (**A**): Conidium of WZ-8A after co-culturing with the bacterial solution of QY43 for 6 h. Conidia that have germinated are indicated by red arrows, while conidia that have not germinated are indicated by blue arrows. Bar = 50 µm; (**B**): Conidial germination rates after 6 h of WZ-8A. The mean ± s.d. (standard deviation) from three independent experiments; ** indicates extremely significant differences at a *p*-value of 0.01; (**C**): Conidium of WZ-8A after co-culturing with the bacterial solution of QY43 for 12 h. The plasmolysis of the WZ-8A conidial is indicated with red arrows. Bar = 10 µm. NB medium without the bacterial solution of QY43 was mixed with the WZ-8A conidial suspension, which was cultured under the same conditions as the CK, the control group. NB medium containing the bacterial solution of QY43 was mixed with the WZ-8A conidial suspension, which was cultured under the same conditions as the treatment group of QY43. NB medium containing the bacterial solution of *E. coli* DH5α was mixed with the WZ-8A conidial suspension, which was cultured under the same conditions as the treatment group of DH5α.

**Figure 3 jof-10-00298-f003:**
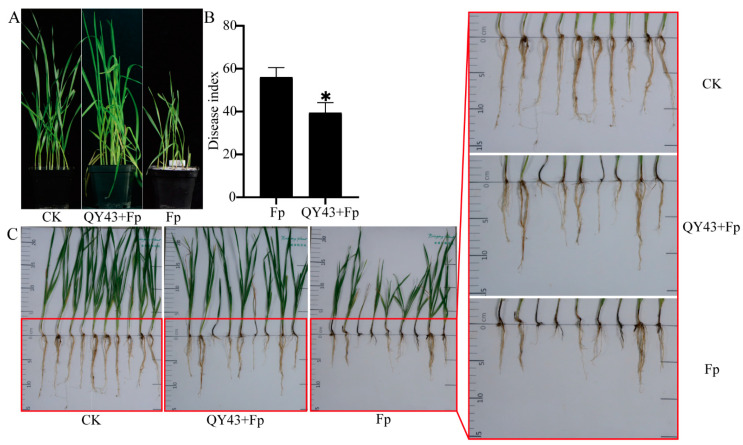
Effect of QY43 in suppressing FCR in a growth chamber. (**A**): Growth of potted wheat was photographed at 21 d; (**B**): The disease index of potted wheat was evaluated at 21 d. The mean ± s.d. (standard deviation) from three independent experiments; * indicates significant differences at a *p*-value of 0.05; (**C**): Plant characteristics and symptoms of FCR under different treatments. The red box indicates that the image is partially enlarged. Millets without *F. pseudograminearum* WZ-8A (Fp) were mixed with the soil as the control group (CK). Millets with WZ-8A were mixed with the soil as Fp. The seeds were soaked for 2 h in the bacterial solution of QY43 with OD600 = 0.8, then, the seeds planted in the soil were mixed with millets containing WZ-8A as QY43 + Fp.

**Figure 4 jof-10-00298-f004:**
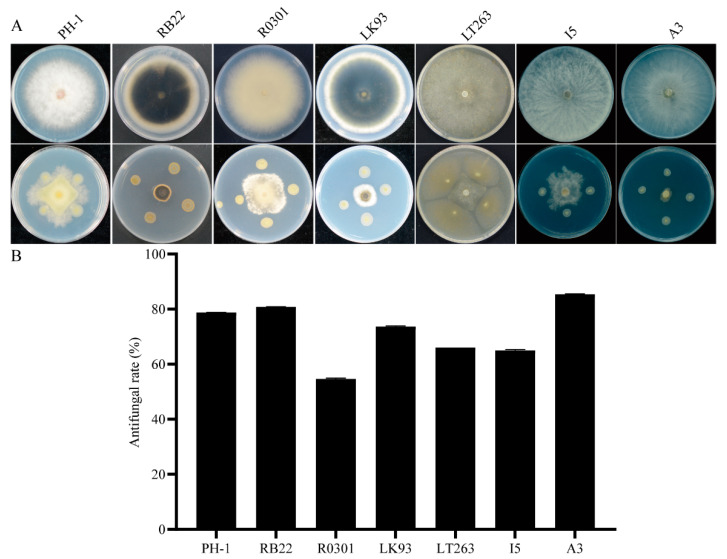
Broad-spectrum antagonistic activity determination of biocontrol bacteria QY43 against different plant pathogens. (**A**): Test of broad-spectrum antagonistic activity of QY43: *F. graminearum* PH-1, *M. oryzae* RB22, *R. zeae* R0301, *B. sorokiniana* LK93, *P. capsici* LT263, *B. dothidea* I5, and *V. mali* A3; (**B**): Antifungal activity rate of QY43. Data are the mean ± s.d. (standard deviation) of three independent experiments. Standard deviation for PH-1, LK93, RB22, LT263, R0301, A3, and I5, respectively, is 0.05%, 0.06%, 0.33%, 0.22%, 0%, 0.29%, and 0.16%.

**Figure 5 jof-10-00298-f005:**
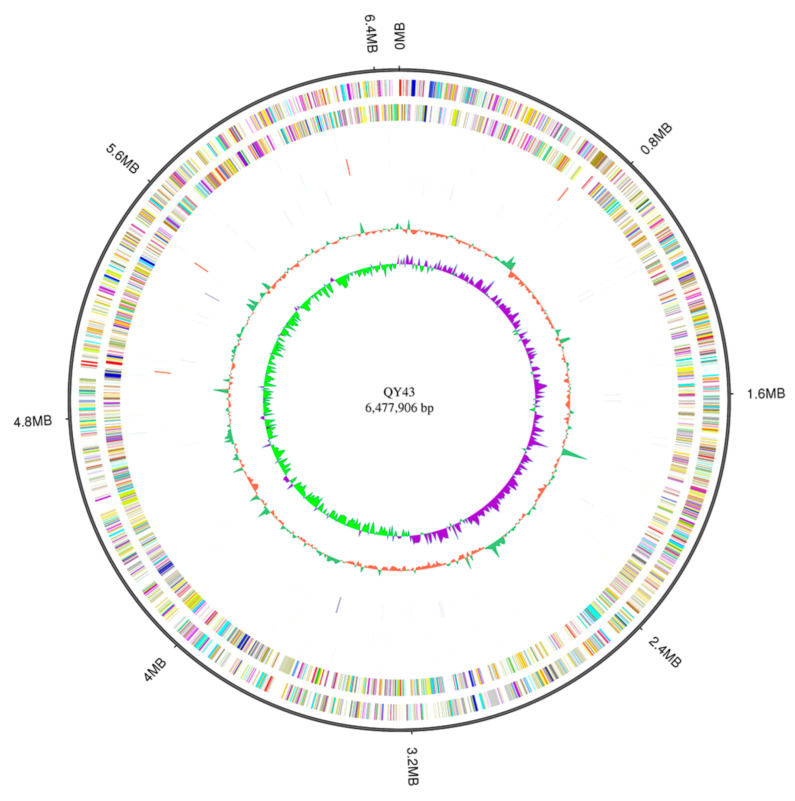
Complete genome circular map of QY43. The Circos diagram was constructed based on 1: genome size; 2: forward strand gene, colored according to the cluster of orthologous group (COG) classification; 3: reverse strand gene, colored according to COG classification; 4: forward strand ncRNA; 5: reverse strand ncRNA; 6: repeat sequences; 7: GC; and 8: GC-skew.

**Figure 6 jof-10-00298-f006:**
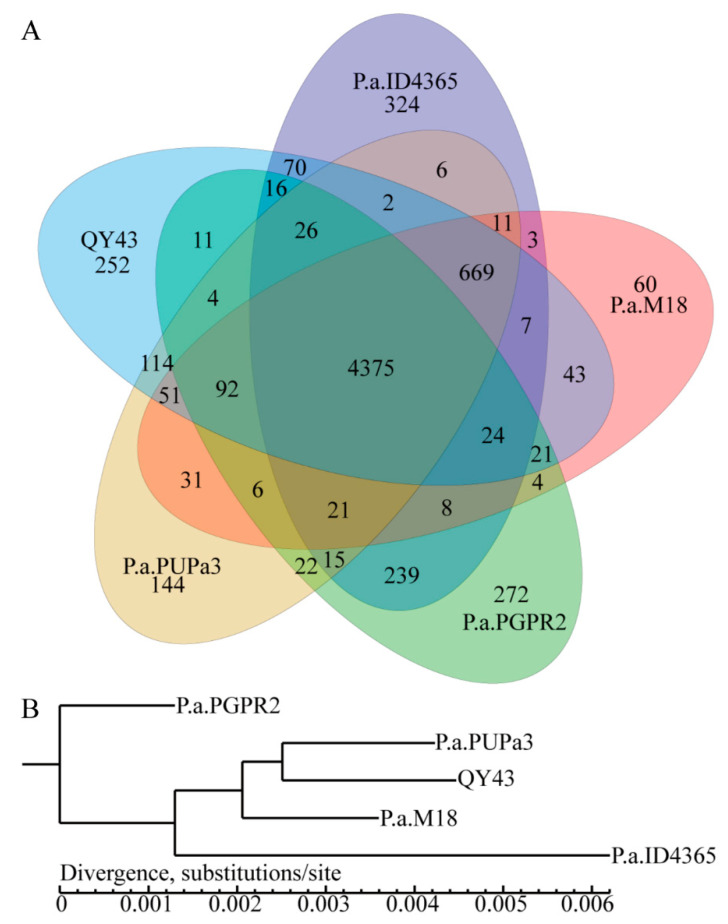
QY43 was compared with the genomes of PUPa3, PGPR2, ID4365, and M18. (**A**): Analysis of shared and unique genes. Each ellipse represents a genome of *P. aeruginosa*, and the data within each region indicate the number of genes that appear exclusively in that specific sample region. A gene with over 50% similarity and a sequence length difference of less than 0.3; (**B**): Phylogenetic tree of ID4365, M18, PGPR2, PUPa3, and QY43, based on results of core gene analysis.

**Figure 7 jof-10-00298-f007:**
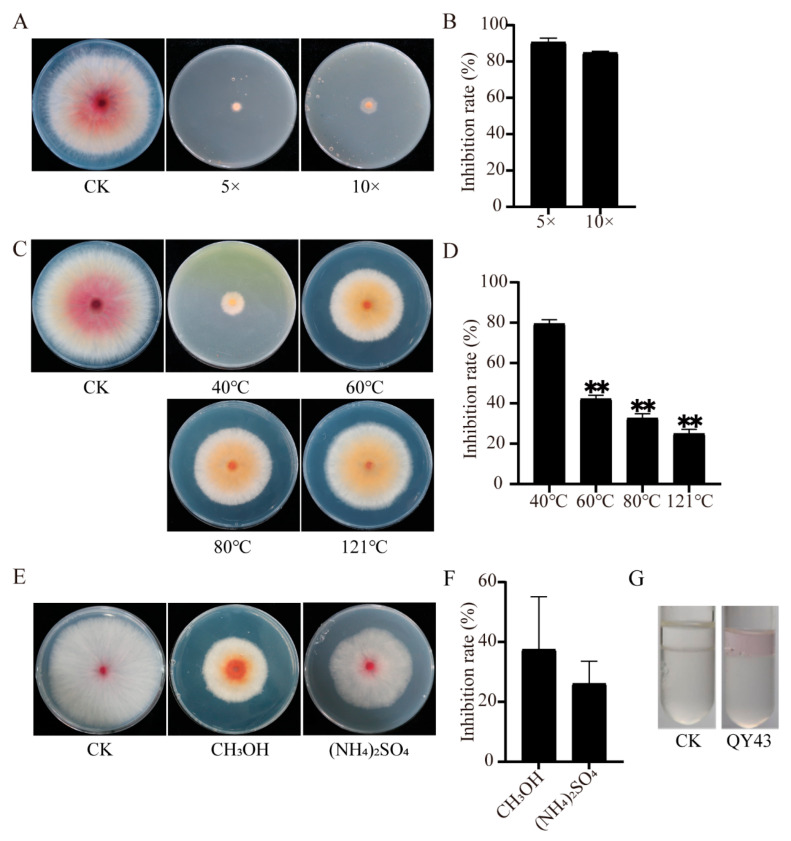
Evaluation antifungal activity of QY43 fermentation broth and identification of antifungal compounds. (**A**,**B**): Antifungal experiment of the different concentrations of QY43 fermentation broth against *F. pseudograminearum* WZ-8A. Data are the mean ± s.d. (standard deviation) of three independent experiments. Normal growing WZ-8A, without experimental treatment, served as CK; (**C**,**D**): Antifungal experiment under a high temperature to determine the tolerance of the QY43 fermentation broth against WZ-8A. The mean ± s.d. of three independent experiments; ** indicates an extremely significant difference at a *p*-value of 0.01. Normal growing WZ-8A, without experimental treatment, served as CK; (**E**,**F**) Antifungal activity of polypeptide extraction from QY43 fermentation broth against it. Data are the mean ± s.d. of three independent experiments. Normal growing WZ-8A, without experimental treatment, served as CK; (**G**): Pyocyanin secretion was detected through the color reaction of QY43. In the solution containing pyocyanin, the upper portion of the solution in the tube will exhibit a red coloration. *E. coli* was cultured under the same conditions as the control (CK). Statistical Product and Service (version 20.0; IBM, Armonk, NY, USA) was used to obtain the standard deviation and the value of the Student’s *t* test.

**Table 1 jof-10-00298-t001:** QY43-specific antifungal-compound-related genes.

Gene ID	Gene Annotation
QY43GL001843	Lipopolysccharide (LPS) O-antigen chain length determinant protein, WzzB/FepE family
QY43GL001844	UDP-N-acetyl-D-mannosaminuronate dehydrogenase
QY43GL004463	sRNA-binding carbon storage regulator CsrA

**Table 2 jof-10-00298-t002:** Biological properties characteristics of QY43.

Strain	Amylase Productions	Cellulase Production	Protease Production	Inorganic Phosphorus Decomposition	Biofilm Formation	Siderophore Production
QY43	−	+	+	+	+	+

## Data Availability

Data are contained within the article or Appendix A. The genome of QY43 has been submitted to the China National Center for Bioinformation, submission number WGS086474.

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
