# Peer review of "Genomics Analysis Reveals the Potential Biocontrol Mechanism of Pseudomonas aeruginosa QY43 against Fusarium pseudograminearum"

_jof, 2024, doi:10.3390/jof10040298_

Round 1

Reviewer 1 Report (Previous Reviewer 1)

Dear Authors:

After having reviewed the manuscript entitled "Genomics Analysis Reveals the Potential Biocontrol Mechanism of Pseudomonas aeruginosa QY43 against Fusarium pseudograminearum", I think that the proposal has a lot of potential for publication, as it reports data that may be interesting for the scientific community interested in the possible use of bacteria and/or their products as biological control agents, as well as plant growth promoters. However, the manuscript has important details that the authors should correct, address or argue in order to suggest its publication. The details to be corrected are below.

Kind regards,

Reviewer´s comments to jof-2956202 manuscript entitled “Genomics Analysis Reveals the Potential Biocontrol Mechanism of Pseudomonas aeruginosa QY43 against Fusarium pseudograminearum

Abstract

Line 14. Add scientific name to wheat (Triticum aestivum) only in line 14…

Introduction

I suggest that the wheat be writing with scientific name abbreviate after mentioning its common and scientific name the first time, throughout the document.

Lines 36 and 37. The authors wrote: Therefore, wheat is one of importantly economical crop in China. This sentence is repeated in line 33, correct.

Line 46. Please, review “pose” word.

Line 66. Change P. fluorescens, by Pseudomonas fluorescens,

Line 68. Change the writing by “dozens of members of the P. aeruginosa species have been isolated and identified...

In the last paragraph of the introduction, the authors don´t mention the aim of the study. Please include it according to the Journal guidelines.

Materials and Methods

Line 90. Change Materials and methods by Materials and Methods

Line 91. In all topics and subtopics, please follow the Journal guidelines, e.g. "2.1. Sample Collection and Bacterial Isolation" should be written.

Line 104. Aikang58 or Aikang 58?

Lines 102 and 103. Please, indicate shortly what kind of molecular analysis was carried out to identify the fungal isolate, e.g. what kind of genes or regions were sequenced for identification and whether a phylogenetic tree was made.

Line 122. I think that the antagonistic activity formula must be antagonistic activity (%) =, not 100%

Line 123. Cake diameter? If the authors are referring to the diameter of the Petri dish, could the authors change this sentence?

Line 125. Add the final period, after USA).

Line 126. If the QY43 strain was identified only using 16S ribosomal DNA. If the authors sequenced the genome of the isolate, what is the point of sequencing the 16S gene separately? Argue.

Line 134. Why OD of 0.1 in QY43? Explain shortly in relation UFC/mL.

Line 137. After water) add a comma: water), with…

Line 139. The conidia concentration wasn´t standardized with a Neubauer chamber?

Line 141. Mixed in a 1:1 relation. 1 x 106 conidia/mL, but how many UFC of bacteria?

Line 143. In what moment the authors decided that the conidia were germinated or not? There are specific methods for this, which one did you take into account? Indicate.

Line 149. Change plugs F. pseudograminearum were, by plugs of F. pseudograminearum were…

Line 151. The authors wrote: steriled soil inoculated, if the soil wasn´t inert, the authors must be included physicochemical analysis of the soil in order to be able to interpret the results obtained.

Line 153. Why OD600 = 0.8 if in the other experiment the OD was 0.1?

Line 154. The authors mention a severity scale from 0 to 7, but do not specify what that scale is, i.e. 0 = healthy plant; 1 = 1-10% of plants diseased; etc. There is also no mention of what the symptoms of the RCF were recorded.

Lines 162-164. In these lines there are scientific names of fungi that are not written in italics. Please correct them. Indicate briefly where these fungal isolates were obtained from.

Line 173: The authors wrote: To identify the antifungal compounds produced by QY43, I suggest writing as follows: To identify the possible antifungal compounds produced by QY43,

Line 198. Change phosphate by Phosphate…

Line 199. What PKO means?

Lines 208 and 209. The authors wrote 1 g L-1 Congo red and 1 mol L-1 NaCl; however, further down the lines they write mg/L (line 495). I suggest homogenizing the units throughout the document according to the Journal guidelines.

Line 235. How was biofilm synthesis specifically measured? By spectrophotometry? Please indicate.

Lines 235 and 236. I suggest join both paragraphs.

Lines 243 and 244. Does the filter have a diameter of 0.22 mm or does the membrane containing the filter have a diameter of 0.22 mm? Argue or correct if necessary.

Line 250. Change 40 , 60 , 80 , and 121 for 20min, respectively., by 40, 60, 80, and 121 for 20 min, respectively.

Line 252. Change 25 for 3 days, by 25 for 3 d,

Lines 260 and 261. How was the methanol evaporated? Rotavapor or not?

Line 261 and 262. original concentration repeats, correct.

Line 266. With which instrument was the sample dialyzed? Please indicate.

Line 268. Polypeptide and peptide are the same? Correct if you consider it necessary.

Lines 270-272. It is not necessary to indicate at the end of each section that the experiments were conducted in 3 independent experiments with 3 replicates, it is sufficient to indicate this in a special statistical analysis section. If the first experiment was only done once because of its complexity, it is sufficient to indicate this at the end of the experiment, but only in that section.

Line 282. Here, I suggest a special section of Statistical analysis.

Results

Lines 295-300. Molecular identification was only based on the 16S gene? If so, it should be treated as partial identification. Although the authors sequenced the entire genome, surely, they could analyze other conserved P. aeruginosa markers to make a complete identification.

Figure 1, S1. What Fp means?

Line 301. Figure caption 1. Subparagraph (d) should be preceded by ;

The letters that make up figure 1 are in upper case, while in the figure caption the letters of the corresponding captions are in lower case. Correct.

Line 302. Change F. pseudograminearum., by Fusarium pseudograminearum.

Line 305. The statistical significance 0.01 it is not mentioned in the materials and methods, so I suggest including a special section indicating the correct level of significance. Is it 0.05 or 0.01? Correct if necessary.

Lines 310-312. This information should be in Materials and Methods section.

Line 315. In vitro in italics.

Line 318. What CK means?

The letters that make up figure 2 are in upper case, while in the figure caption the letters of the corresponding captions are in lower case. Correct.

Figure 2. Honestly, I see no difference between figure 1 with CK and without CK at 6 h after treatment. If the authors observe differences or abnormalities at 6 hours, please clearly indicate this in the panel (arrows). If the authors observed plasmolysis at 12 h, I would choose only that evidence. Often, in the figures, it looks like 6h or 12h, shouldn't it be 6 h and 12 h? Please check and correct if necessary. Figure 2 B. How did you do a statistical analysis of 3 treatments? Student T is done to compare two treatments. If the authors compared 3 treatments, they should have done an ANOVA and then a Tukey's test. 0.01 significance level should be stated in the materials and methods. No effect on DH5a? How does it differ from the treatment of fungal conidia? I see no difference, please state this clearly in the text. Were the data for inhibition percentages or rates transformed to use the T Student or is it not necessary? Please argue.

Line 325. Change µm;(b):, by µm; (b):

Line 345. Change (p ˂ 0.05) by (p ˂ 0.05).

Figure 3. What are the symptoms of FCR on the plants? I don't see any symptoms on the plants in figure 3A, I only see large plants and smaller plants. I suggest that the authors look for and place plants representative of what they observed and measured. Figure 3B. I suggest change Disease index by Disease index (%). Although the disease index was significantly reduced by the treatment, the authors consider that this reduction would be sufficient and would benefit wheat production in the field? It would have been important to introduce a fungicide control to compare the effect of their bacterial treatment with the latter. The evidence of the roots is perfect, I would go for these, but this is just one comment you can leave out. I suggest that the results be explained a bit more in the text, that the authors address the root data a bit more, as only the overall effect on FCR is mentioned.

Lines 346 and 347. This information should be in Materials and Methods section.

Lines 360 and 361. I suggest delete the standard deviations of the text, finally it is mentioned in the figure caption.

Line 361. I suggest that the fungi be mentioned in the text in the same order in which they are arranged in figure 4A. Again, the letters in the figures are in capitals and in the figure caption they are in lower case, please correct.

Lines 370 and 371. I don't understand the statistical information, did you compare the effect of the bacterial extract on the fungi with a control? If yes, what was the control? If the authors compared all treatments against any control or against each other, they should have done ANOVA and TUKEY no T Student.

Figure S2. I suggest homogenizing the size of the letters in the captions, i.e. that they should be in upper or lower case in all the figures in the manuscript.

Line 378. The content of GC corresponded to P. aeruginosa?

Line 381. Change Figure 5.The, by Figure 5. The

The title of Table S1 must have a full stop.

Line 423. Table 1 should appear first in the text rather than Figure 7.

Figure 7D. The same comment of the statistical analysis.

Line 433. Change (Figure 7g).Based by (Figure 7g). Based… Change de writing in this sentence: QY43, QY43 was established, Trying not to repeat QY3.

Line 436. The authors wrote: Thus, QY43 has practical agricultural applications as a biological control agent. Change the writing like this: Thus, QY43 could has diverse practical agricultural applications as a biological control agent and biofertilizer.

Discussion

Lines 458-460. The production of pyocyanins and siderophores suggests that the QY43 isolate has growth-promoting capacity as a direct or indirect mechanism, please clarify in the text.

Lines 461 and 462. The authors wrote: The biocontrol factors include various antifungal compounds and substances that promote plant growth. Clarify, as direct or indirect mechanism.

Conclusions

I don’t have any comment in this section.

References

I suggest that authors stick to the Journal's guidelines for writing references, as I observe heterogeneity and errors in some of them.

Author Response

Reviewer 2 Report (Previous Reviewer 3)

Page 4, Line - 161 - 164

Scientific names need to be italicized.

Page 5, Line - 197

Subtitle - 2.8 Identification of physical and chemical properties of QY 43 is not appropriate. QY 43 is not a chemical compound or a physical material. 

Page 10, Line 390 -391 and Page 13, Line 503 - 504 - reference to two genes phoQ and phoP requires serious reconsideration. Are you sure that those genes are involved in bacterial P uptake not in signaling system? Also, how does bacterial P uptake/metabolism contribute to plant growth (citation)?

Please comments

Author Response

This manuscript is a resubmission of an earlier submission. The following is a list of the peer review reports and author responses from that submission.

Round 1

Reviewer 1 Report

Comments and Suggestions for Authors

Dear Athors,

The manuscript entitled "Genomics Analysis Reveals the Biocontrol Mechanism of Pseudomonas aeruginosa QY43 against Fusarium pseudograminearum" shows very interesting results for me and I think also for the scientific community, since important data are presented on an isolate of Pseudomonas aeruginosa (QY43) with diverse characteristics, including the biocontrol of the fungus Fusarium pseudograminearum. What I like is the genomic approach. However, I think that in some sections the information is disorganized, unclear and therefore confusing. I think the manuscript is worthwhile and could be publishable, but it needs more organization and clarity from the authors. Below are some specific comments on the manuscript that your should address.

Best regards,

Reviewer´s comments to Journal of Fungi 2848051 manuscript titled “Genomics Analysis Reveals the Biocontrol Mechanism of Pseudomonas aeruginosa QY43 against Fusarium pseudograminearum

Title

The title should be rewritten, as the authors did not provide strong or clear data on the biocontrol mechanism of the Pseudomonas aeruginosa QY43 isolate on Fusarium pseudograminearum.

Abstract

Line 24. Change Comparative genomic analysis by comparative genomic analysis…

 Introduction

The introduction seems to me to be well written and well-founded, I only observe three minor details:

1.- There are minimal errors like the one in line 70, in which Trichoderma harzianum (italics) should be abbreviated as T. harzianum (italics).

2.- I suggest to add briefly the nutritional and economic importance of wheat in China.

3.- The objective statement is not clear to me, it does not coincide with the title of the manuscript, I suggest that it be analyzed and rewritten.

 Material and Methods

Line 105. Change Tryptic Soy Broth by tryptic soy broth…

Line 107. Change 25°C by 25 °C in all document.

Lines 110 and 111. Respect to F. pseudograminearum wild-type strain WZ-8A, I suggest to indicate briefly how was the fungus isolated and identified.

Line 117. How was the antagonistic effect obtained or measured?

Lines 118 and 119. The authors wrote: Subsequently, two drops (1 μL) of bacterial solution of QY43 cell suspension, cultured for 24 h in nutrient broth (5 g peptone, 3 g beef extract, 5 g NaCl, and 1 L of distilled water; pH = 7.4 ± 0.1), were deposited around the plug, approximately 2.5 cm away. How many CFU/mL did this inoculum have? This determination is important, since the concentration of possible antifungal compounds may depend on the bacterial concentration used. Why QY43 if several antagonistic bacteria were mentioned before? Was this experiment performed to confirm the antifungal effect or what was the purpose of this experiment?

Line 119. Nutrient broth does not have abbreviations such as PDA?

Line 123. The authors wrote: all isolates were assessed in triplicate. Did the authors perform at least two independent experiments?

Line 134. I think that this section 2.3. Assessment of Broad-Spectrum Antagonistic Activity of QY43 must be after 2.5. Efficacy of QY43 Against FCR in A Growth Chamber section.

Line 136. Phytophthora capsica LT263 or Phytophthora capsici LT263…

Line 138. Change Phytophthora capsici by P. capsici (italics).

Lines 143 and 144. The authors wrote: all isolates were assessed in triplicate. Did the authors perform at least two independent experiments?

Line 147. What does NB mean?

Line 148. Why value of 0.1?

Lines 152 and 153. The authors wrote: Adjusted concentrations of the QY43 bacterial solution and WZ-8A conidial suspension were mixed in a 1:1 ratio. Which was the bacterial concentration?

Line 154. Change Escherichia coli by Escherichia coli (italics)…

Line 155 and 156. The authors wrote: the percentage of 6-h germinated conidia. How the authors considered when conidia germinated and when not germinated? Indicate.

Line 164. OD600 or OD 600?

Line 165. How was the disease index documented? Indicate.

Line 178 and 193. Change P. aeruginosa by P. aeruginosa (italics).

Line 199. 2.8. Identification of Physical and Chemical Properties of QY43, for what? For growth promotion? Indicate.

Line 204, 209, 216, 222, 228 and 231. Change 30°C by 30 °C.

Lines 210 and 211. 1 gL-1 and 1 mol L-1, these units match the Journal guidelines? Please check.

Line 226. What CAS mean?

Lines 243 and 235. How was biofilm formation quantitatively determined?

Phosphate solubilization test, Cellulose degradation ability test, Protease secretion ability test, Amylase secretion ability test, Siderophore secretion ability test, Biofilm formation ability test. Shouldn't all these techniques carry author citations? No, argue, yes, argue.

Line 238. One milliliter or 1 mL, homogenize.

Line 247, 253, 254, 255. Change 4°C by 4 °C.

Line 248. Change 45°C by 45 °C.

Line 250. What PBS means?

Lines 250 and 251; 257. The authors wrote 0.22 μm bacterial filter: Filter or membrane?

261 and 262. The authors wrote: The results of the genomic analysis indicated that QY43 may produce pyocyanin; therefore, we determined whether QY43 could produce pyocyanin. This text should not go in this section; it should be changed to results.

Line 263. What NA liquid medium means?

Lines 265 and 266. The authors wrote: described, described by who?

Line 236. 2.9. Characterization of Antifungal Compounds Produced by QY43, this section does not contain citations of authors, why?

Results

The way the authors write the results is very confusing to me, as they describe numerous data and discuss them, for example, from line 418 to 426. This section should only contain data obtained from the experiments, not their discussion. This makes the discussion section very poor for the amount of data obtained. There needs to be better organization in the way the results are presented, it is confusing to me.

Unfortunately, I did not have access to Figure S1.

Line 291. Change P. aeruginosa by P. aeruginosa (italics).

Line 291 and 292. The authors wrote: Consequently, QY43 was confirmed to be P. aeruginosa. From a taxonomic point of view, is the sequencing of a single molecular marker, in this case the 16S rRNA partial gene, sufficient to consider the isolate as a strain of P. aeruginosa?

Line 297. What s.d. means? Here, the authors wrote: from three independent experiments; In the materials and methods section, the authors just mentioned that they used three repetitions, not independent experiments, but here do it. Please clarify.

Lines 297 and 298. The authors wrote: ** indicates significant differences at a p-value of 0.01. The experimental design used and the statistical analysis should appear in the materials and methods section. The software used should also appear. All this is absent from this section.

I additionally suggest that the caption of Figure 2 be modified, and that the names of the inhibited fungi be in the same order as in the figure.

Line 305. Change Phytophthora capsici by P. capsici (italics)

Lines 307 and 308. How were the percentages of antagonistic activities obtained?

Line 309-311. You cannot conclude in this section, please change the sentence.

Line 316. s.d. of three independent experiments. What s.d. means? Here, the authors wrote: from three independent experiments; In the materials and methods section, the authors just mentioned that they used three repetitions, not independent experiments, but here do it. Please clarify.

Line 319. Check if the Journal guidelines allow writing in vitro without italics.

Line 320. mycelial growth??? Check.

Figure 3. The letters of the figures in the text must be the same as the letters of the figures. Here, the text uses lowercase letters and the figure includes uppercase letters. That confuses me.

Figure 3B. What CK means? The result of the graphic 3B is very clear, but the photographs CK+Fp, QY43+Fp and DH5a+Fp (6 and 12 h) show nothing. If the authors want me to see any distinctive features in the photographs, they could use arrows.

Line 324. The authors wrote: abnormalities, could the authors please tell me which abnormalities?

Line 331. The authors wrote: The disease index was evaluated. How the authors obtained the disease index?

Line 333: The authors wrote: pseudograminearum (p < 0.05). Again, statistical analysis is not shown. It should be shown in the corresponding section.

Figure 4A. What CK means? In this figure, treatment Fp, sincerely, I can´t saw the disease in wheat plants. I only see high plants and low plants. What are the symptoms of FCR for comparison? It would be nice if the authors could show clearer pictures showing a control with clear symptoms of FCR.

Line 334. FCR or Fusarium crown rot? Homogenize.

Unfortunately, also I did not have access to Figure S2A and S2B.

Justify the foot of figure 5.

Line 360. Change Chlamydia[43].,by Chlamydia [43].

Unfortunately, also I did not have access to Table S1.

Line 371. What phz-related means?

Line 385. What rhl-related means?

Lines 397 and 398. Change si-derophore[49]; by si-derophore [49];

Unfortunately, also I did not have access to Figure S3a-d.

Line 425. Change P. aeruginosa by P. aeruginosa (italics).

Line 442. The effect of temperatures in antagonistic affects were described in material and methods section?

Line 457. Bacteriostatic experiment? Argue why?

Discussion

It seems to me that there should be more clarity on the relationship between the genes associated with biocontrol and the evidence that the authors found on the products of these genes and their relationship with the phenotype of the bacteria QY53. The way this is written is confusing to me, I think they should organize and clarify this section more.

Line 484. Delete the comma after pathogenicity…

Line 487. Change Pseudomonas aeruginosa by P. aeruginosa (italics)

Lines 508-514. This paragraph adds nothing to the discussion of the results of the manuscript. Delete.

Conclusions

I consider that the conclusion falls short of the results obtained in this research work. Additionally, it does not coincide at all with the title of the manuscript, since it talks about a biocontrol mechanism that was not really clearly demonstrated, from my point of view, or the authors did not write it in a clear way. The conclusion should be improved.

References

There are some errors in the references:

1.- Repeated numbering.

2.- The names of the journals should be abbreviated. Consult the Journal guidelines.

3.- Review journal guidelines when writing two author citations together.

Reviewer 2 Report

Comments and Suggestions for Authors

The authors present a manuscript about the use of genomics analysis to detect the biocontrol mechanism of Pseudomonas aeruginosa QY43 against Fusarium pseudograminearum.

Apart from the Introduction which reports the state of the art about the topic proposed, the rest of the manuscript has serious flaws and cannot be accepted for publication.

The M&Ms are poor in details and information useful for repeatability. The experimental scheme is not clear and the authors do not report any information about the bioinformatic that should be the core of the paper (i.e. platform used for sequencing, bioinformatic analysis, and tools or pipelines). Moreover, with few exceptions, no replicates are reported. Statistic analysis is not mentioned. As a consequence, results are confused and not adequately supported by the literature.

Comments on the Quality of English Language

Moderate editing of English language required

Reviewer 3 Report

Eco-friendly, cost-effective alternatives to plant fungal disease control are needed to as a replacement or complement for the chemical fungicides. Microbial-based biocontrol strategies are emerging as promising alternatives to the use of synthetic fungicides. Pseudomonas spp. may be good candidates for biocontrol agents due to its colonizing ability, inducing plant systemic resistance and to produce antifungal compounds.  The manuscript titled – Genomics analysis reveals the biocontrol mechanism of Pseudomonas aeruginosa QY43 against Fusarium pseudograminearum by Jiaxing Meng et al. describes isolation and characterization of bacterial isolate QY43 (Pseudomonas aeruginosa) from soil, its antagonistic activity against fungal pathogens including F. pseudograminearum (wheat fusarium crown rot pathogen), its antifungal compounds, evaluation of QY43 treatments for controlling wheat fusarium crown rot under growth chamber conditions, and whole-genome sequencing and comparative genomic analysis of QY43.

The title does not reflect the work/results, authors should consider using "potential" before the biocontrol mechanisms.

Page 1, Line 16 – 18.  

Abstract: In this study, we isolated 37 strains of biocontrol bacteria displaying antagonistic effects against F. pseudograminearum WZ-8A from over 8000 bacteria isolated from soil samples from wheat fields with a high incidence of FCR.

The statement “8000 bacteria isolated” is repeated also in the discussion section.

Page 6, Line 278 -279

Results: A total of 37 isolates were assessed for antagonistic activity against F. pseudograminearum WZ-8A (Figure S1).

The details regarding “8000 bacterial isolated” needs to be included in the results section.

Page 3, Line 103

Soil samples were collected from a high-incidence area of FCR.

There is not enough information regarding soil sampling - where exactly the soil samples were collected, Geographical location (GPS coordinates), what type of field - cropped or uncropped, what crop, rhizosphere soil or bulk soil, during what season, soil sample depth, how the samples were collected etc. All these are very pertinent to the study.

Page 3, Line 163 - 165

The seeds were soaked for 2 h in a QY43 bacterial solution with OD600 = 0.8. Observations of wheat growth were recorded at 21 d, and the disease index was documented.

Provide citations for the rating system that was used for assessing the disease index.

Were there any differences in the visual discoloration of sub-crown internode/leaf sheath between the treatments?

Page 8, Line 316.  

Figure 2. (b): Broad-spectrum antifungal activity rate of QY43. Data are the mean ± s.d. of three independent experiments.

No clear information available within the methods section describing how the antifungal activity rate was determined.

Page 10, Line 390 -391

Moreover, QY43 had numerous genomic signatures that favor plant growth, including two genes involved in phosphate metabolism.

Which specific genes? How does the genes that are involved in bacterial internal phosphate metabolism favor plant growth?

Page 11, Line 432 – 434

To delve deeper into the biocontrol mechanism of QY43, we assessed QY43 for the secretion of protease, cellulase, amylase, and siderophores, and analyzed organic phosphorus decomposition and biofilm formation.

Provide evidence (citation) for involvement of organic phosphorus decomposition in biocontrol mechanism.

Page 13, Line 485 - 486

Although this study did not specifically investigate the enhancement of crop resistance as a biocontrol measure

Crop resistance is not considered as a type of biocontrol.

Page 13, Line 503 - 504

but also possessing genes associated with plant growth promotion.

Which specific genes?